# Neuroprotection of Cholinergic Neurons with a Tau Aggregation Inhibitor and Rivastigmine in an Alzheimer’s-like Tauopathy Mouse Model

**DOI:** 10.3390/cells13070642

**Published:** 2024-04-06

**Authors:** Maciej Zadrozny, Patrycja Drapich, Anna Gasiorowska-Bien, Wiktor Niewiadomski, Charles R. Harrington, Claude M. Wischik, Gernot Riedel, Grazyna Niewiadomska

**Affiliations:** 1Mossakowski Medical Research Institute, 02-106 Warsaw, Poland; mzadrozny@imdik.pan.pl (M.Z.); pdrapich@imdik.pan.pl (P.D.); agasiorowska@imdik.pan.pl (A.G.-B.); wniewiadomski@imdik.pan.pl (W.N.); 2School of Medicine, Medical Sciences & Nutrition, University of Aberdeen, Aberdeen AB25 2ZD, UK; c.harrington@abdn.ac.uk (C.R.H.); cmw@taurx.com (C.M.W.); g.riedel@abdn.ac.uk (G.R.); 3TauRx Therapeutics Ltd., Aberdeen AB24 3FX, UK; 4Nencki Institute of Experimental Biology, 02-093 Warsaw, Poland

**Keywords:** cholinergic system, tauopathy, Alzheimer’s disease, hydromethylthionine, acetylcholinesterase inhibitors, rivastigmine, symptomatic treatment

## Abstract

Basal forebrain cholinergic dysfunction, most likely linked with tau protein aggregation, is a characteristic feature of Alzheimer’s disease (AD). Recent evidence suggests that tau protein is a putative target for the treatment of dementia, and the tau aggregation inhibitor, hydromethylthionine mesylate (HMTM), has emerged as a potential disease-modifying treatment. However, its efficacy was diminished in patients already receiving approved acetylcholinesterase inhibitors. In this study, we ask whether this negative interaction can also be mimicked in experimental tau models of AD and whether the underlying mechanism can be understood. From a previous age profiling study, 6-month-old line 1 (L1) tau transgenic mice were characterized by a severe reduction in several cholinergic markers. We therefore assessed whether long-term pre-exposure with the acetylcholinesterase inhibitor rivastigmine alone and in conjunction with the tau aggregation inhibitor HMTM can reverse cholinergic deficits in L1. Rivastigmine and HMTM, and combinations of the two compounds were administered orally for 11 weeks to both L1 and wild-type mice. The brains were sectioned with a focus on the basal forebrain, motor cortex and hippocampus. Immunohistochemical staining and quantification of choline acetyltransferase (ChAT), tyrosine kinase A (TrkA)-positive neurons and relative optical intensity (ROI) for vesicular acetylcholine transporter (VAChT), and acetylcholinesterase (AChE) reactivity confirmed reversal of the diminished cholinergic phenotype of interneurons (nucleus accumbens, striatum) and projection neurons (medial septum, nucleus basalis magnocellularis) by HMTM, to a greater extent than by rivastigmine alone in L1 mice. Combined administration did not yield additivity but, in most proxies, led to antagonistic effects in which rivastigmine decreased the benefits shown with HMTM alone. Local markers (VAChT and AChE) in target structures of the basal forebrain, motor cortex and hippocampal CA3 seemed to be normalized by HMTM, but not by rivastigmine or the combination of both drugs. HMTM, which was developed as a tau aggregation inhibitor, strongly decreased the tau load in L1 mice, however, not in combination with rivastigmine. Taken together, these data confirm a cholinergic phenotype in L1 tau transgenic mice that resembles the deficits observed in AD patients. This phenotype is reversible by HMTM, but at the same time appears to be subject to a homeostatic regulation induced by chronic pre-treatment with an acetylcholinesterase inhibitor, which interferes with the efficacy of HMTM. The strongest phenotypic reversal coincided with a normalization of the tau load in the cortex and hippocampus of L1, suggesting that tau accumulation underpins the loss of cholinergic markers in the basal forebrain and its projection targets.

## 1. Introduction

Until recently, the only approved drugs for AD have been anticholinesterase inhibitors and memantine. Disease-modifying treatments are being sought that can remove or prevent the formation of the characteristic pathologies of AD, namely β-amyloid plaques and neurofibrillary tau tangles. Two amyloid immunotherapeutic drugs, aducanumab and lecanemab, have been found to produce small benefits for patients in several Alzheimer cohorts [1,2,3,4] and reignited the search for novel treatments. A tau aggregation inhibitor, hydromethylthionine, typically administered as a mesylate salt (HMTM), has been developed to target abnormal tau processing, and an exploratory analysis confirmed the high efficacy of HMTM when administered to a drug-naive patient sub-cohort [5,6]. Since the majority of recruits, however, were already receiving symptomatic therapies (acetylcholinesterase inhibitors or memantine), the HMTM drug effect appeared to be lost due to a negative interference between the therapies. Here, we attempted to (i) back-translate the human treatment regime to the mouse model and (ii) get a richer understanding of the potential mechanisms underlying this interference effect.

Of particular interest was the relationship between the cholinergic system, tau aggregation and cell loss. In the nucleus basalis of Meynert (NBM), cytoskeletal changes are observed from the early Braak stages of AD based on tau pathology [7] and more than 50% of neurons in NBM degenerate in supranuclear palsy due to tau aggregation [8]. These results are consistent with our own observations of tau transgenic animals in which there is considerable degeneration of basal forebrain cholinergic neurons [9], and strongly support the notion that the onset of tau aggregation in the basal forebrain plays a significant role during the emergence and progression of AD [10,11,12,13]. Due to the selective requirement of the cholinergic system for trophic support [14], this may also explain its excessive susceptibility to the propagation of tauopathies. This would lead to a continuous decrease in the concentration of the neurotransmitter acetylcholine, a phenotype that can be rescued in transgenic mice by cholinesterase inhibition and HMTM [15], concomitant with a reversal of learning and memory deficits [12,16,17].

In the Braak staging model [7,18], cholinergic projection areas (isocortex, hippocampus) are particularly vulnerable to tau aggregation and spreading [19,20,21]. The reversal of tau aggregation, particularly in basal forebrain cholinergic neurons and then by their tonic release in the hippocampus and cortex, would seem a viable strategy for the development of new therapeutics, including HMTM [6,22]. The HMTM-dependent prevention of tau self-assembly may be effective in slowing the progression of AD and other tauopathies [23], halting the emergence of pathological tau deposits due to the collapse of the intracellular and axonal cytoskeleton [24]. The failure of tau to bind to microtubules and support their assembly, stabilization and spacing, axonal length and diameter, and provide neuronal rigidity results in neurotoxicity to multiple transmitter systems [25], including cholinergic basal forebrain neurons [26,27,28].

Hydromethylthionine mesylate (HMTM), a stable reduced form of methylthioninium, has been shown to act as a selective inhibitor of tau protein aggregation in cell-free systems and in cellular and murine models of tauopathy [29,30,31]. An oxidized form, methylthioninium chloride (MTC), when administered as a monotherapy in patients with AD, was also effective in alleviating symptoms of the disease [32]. In two independent phase III studies, however, HMTM administered to patients with mild or moderate AD was safe and effective as a monotherapy, but not when administered as an add-on to symptomatic treatments (e.g., donepezil, rivastigmine and memantine) [5,33].

Therefore, we sought to mimic the latter clinical trial designs pre-clinically in the line 1 (L1) AD mouse model. L1 mice present with the typical Braak-like spreading of the tau pathology and manifest cognitive impairments [30] at 5–6 months of age, together with minimal sensorimotor deficits and significant loss of cholinergic staining in the basal forebrain, cortex and hippocampus [9]. While cognitive function can be rescued with HMTM monotherapy, there is strong interference when acetylcholinesterase inhibitors (i.e., rivastigmine) are pre-administered before HMTM [34]. Here, we assessed the consequences of the single and combined administration of rivastigmine and HMTM for neuropathological endpoints related to the cholinergic system and tau aggregation.

## 2. Materials and Methods

### 2.1. Animals and Study Design

Experiments were carried out in accordance with the European Communities Council Directive (63/2010/EU), a project license with ethical approval from First Warsaw Local Ethics Committee for Animal Experimentation, and were carried out in accordance with the Polish Law on the Protection of Animals and National Institute of Health’s Guide for Care and Use of Laboratory Animals (publication no. 85-23, revised 1985).

Experiments were conducted on female transgenic L1 mice and wild-type controls, Naval Medical Research Institute mice (NMRI) mice. L1 was generated via the insertion of the plasmid pSS296-390 containing a human tau fragment coding for the amino acid sequence 296–390 of the largest CNS tau isoform, htau40 [35], fused with a signal-sequence-directing protein to the endoplasmic reticulum [30]. The transgenic line is characterized by the expression of a truncated form of tau in the brain which emulates the spread of tauopathy as observed in -Braak staging. The expression of tau296–390 causes aggregation of tau protein and the deposition of filamentous structures resembling the core fragment of the paired helical filaments (PHFs) that are the principal constituent of neurofibrillary tangles (NFTs). L1 mice also present with cognitive impairment [30].

Mice were bred commercially (Charles River, Margate, UK) in multi-cage isolators on positive pressure and under specific pathogen-free (SPF) conditions. After weaning, they were maintained in isolators in shoe boxes with siblings and aged until appropriate for the experiment with ad libitum access to water and food chow. For acclimatization, the mice were delivered (air and truck) to the Animal Facility of the Nencki Institute (Warsaw, Poland) one month before the start of the experiment. They were housed, according to the genotype, in small colonies of up to 5 mice in a single cage (Type III, 382 × 220 mm). The cage lining included corn cobs, strips of paper, and cardboard tubes as an enrichment. The holding rooms were maintained at a constant temperature (20–22 °C), humidity (60–65%) and the air exchange rate (17–20 changes/h), with a 12 h light/dark cycle (lights on at 6 a.m.). The animals had free access to food and water. In alignment with previous studies, female mice were 19 to 27 weeks old at the start of the experiments [36]. Animals were assigned to treatment groups based on their body weights determined prior to drug administration to have an equal mean body weight for each treatment cohort. In addition, the body weights were recorded weekly during the in-life phase of dosing, and on the final day prior to necropsy. After each drug dosing, the animals were observed for adverse effects for up to 4 h. Clinical signs were recorded and, where possible, assessed for severity by non-invasive screening [37]. Three mice were terminated prematurely due to poor tolerance of the gavaging process/weight loss. No other clinical signs were observed in the remaining animals. Additional behavioral observations were performed during the first two weeks of drug administration in both the first and second phases of the study (Figure 1). This included an assessment of motor activity, changes in reactivity, hind limb standing, freezing and altered social behavior (isolation/dominance) [37]. Transient behavioral anomalies were noted for both HMTM and rivastigmine administrations. Their frequency was irregular and very rare.

Thirteen groups of animals participated in the entire experiment and a total of 116 mice were used. The details are summarized in Table 1. Eight core groups included both L1 and wild-type genotypes with a matching drug administration route and dosing regimen. The drugs included vehicle, HMTM (15 mg/kg) rivastigmine (0.5 mg/kg) and a combination of rivastigmine first (0.5 mg/kg) and HMTM as an add-on (15 mg/kg). All drugs were administered orally by gavage. Five further complementary groups included L1 mice only in order to gain a better understanding of drug actions and the dose–response profiles.

### 2.2. Drugs and Administration

Two substances were tested in this experiment: rivastigmine, an acetylcholinesterase inhibitor for symptomatic therapy in AD [38]; and hydromethylthionine mesylate (N3,N7,N7′,N7′-tetramethyl-10*H*-phenothiazine-3,7-diaminium bis(methanesulfonate), HMTM, also formerly termed leucomethylthionium bis(methanesulfonate), LMTM). HMTM was supplied by TauRx Therapeutics Ltd., Aberdeen, UK, and rivastigmine was purchased from Tocris Bioscience (Bristol, UK, #4440, lot numbers 1A/198244, 1/a206698, 1A/206690). The treatment regime is depicted in Figure 1. The effects of HMTM administration on cholinergic function in L1 tau mice and the interference of HMTM with concomitantly administered rivastigmine were investigated in an 11-week study of two phases. During phase 1 (weeks 1–5), mice received either the vehicle or rivastigmine and, during phase 2 (weeks 6–11), either the vehicle or HMTM were added to the earlier treatment. The exact drugs and doses and group sizes are indicated in Table 1. The doses of rivastigmine were 0.1 and 0.5 mg/kg, while HMTM was given in at 5 and 15 mg/kg.

The oral route was chosen to be consistent with the anticipated clinical route of administration. The drugs were administered via oral gavage at a dose volume of 5 mL/kg of body weight daily for 5 days (Monday–Friday) per week (55 days in total) in the morning between 8 and 10 a.m. Nitrogen-sparged deionized water was used as a vehicle for both drugs. HMTM formulations were prepared fresh on each day of dosing and stored in an amber glass bottle for light protection. Rivastigmine aliquots were used for 2–3 weeks and stored at 4 °C. The HMTM administration schedule and dose were selected based on previous data showing that the tau pathology was successfully reduced and the behavioral phenotype was reversed in tau transgenic mice treated with the item [36]. Rivastigmine doses were chosen based on their effects on extracellular acetylcholine concentration and acetylcholinesterase activity in the rat cortex [39,40].

### 2.3. Animal Sacrifice and Brain Tissue Collection

The day after receiving their last dose of compounds, the animals were transported to the treatment room one by one, weighed and terminally anesthetized via an intraperitoneal injection of Morbital^®^ (sodium pentobarbital, 150 mg/kg). Deep anesthesia was confirmed as the absence of corneal reflex, pain sensation and unconditional reflexes. Anaesthetized mice were transcardially perfused with 30 mL PBS (0.1 M, pH 7.4) containing heparin (0.1 mL of heparin solution (WZF 5000 IU/mL, Polfa Warszawa SA, Warsaw, Poland) for 100 mL PBS), followed by perfusion with 50 mL of 4% paraformaldehyde with 15% saturated picric acid in 0.1 M PBS pH 7.4 and another 30 mL of 5% glycerol with 2% dimethyl sulfoxide (DMSO) in 0.1 M PBS (pH 7.4). Then, the brains were removed and each hemisphere was stored separately at 4 °C for 24 h in 4% paraformaldehyde for post-fixation and tissue penetration. This was followed by incubation in glycerol with DMSO at two different concentrations (10% glycerol for 24 h and then 20% glycerol for 24 h; both solutions containing 0.1 M PBS with 2% DMSO) for further cryo-protection. The hemispheres were then stored at −80 °C for later analysis.

### 2.4. Immunohistochemistry for ChAT and TrkA in Frozen Brain Sections

Due to different histochemical staining protocols, the animals in each experimental group were divided into two cohorts, one for assays on the frozen sections (n = 5) and the other for assays on the paraffin sections (n = 3 to 5). The right brain hemispheres of five randomly selected mice from each experimental group were used to perform ChAT and TrkA immunohistochemistry. The hemispheres were covered with a medium (Cryomatrix^TM^, Thermo Scientific, Rugby, UK), then sectioned coronally (Leica CM1850 freezing microtome, Germany) at −20 °C, a 40 µm thickness and the desired brain levels in accordance with the Mouse Brain Stereotaxic Atlas [41]. Five series of consecutive sections were collected throughout the basal forebrain of each mouse. From these series, one set was processed for ChAT immunoreactivity (ChAT-ir), the second set for TrkA immunoreactivity (TrkA-ir), the third one was stained with Cresyl-Violet (Nissl-staining) for the histological identification of anatomical regions, and the other two sets of sections were stored at −20 °C in a cryoprotectant. For an immunohistochemical analysis, free-floating sections were rinsed three times for 5 min with 0.1 M PBS. Then, the sections were incubated with 1% H_2_O_2_ solution in 0.1 M PBS to block the endogenous peroxidase activity. Next, the tissue material was rinsed three times with 0.1 M PBS and incubated for one hour at room temperature (RT) with a 5% normal goat serum(NGS, for ChAT) or normal rabbit serum (NRS, for TrkA) solution (Vector Laboratories, Cat. No. S-1000-20 and S-5000-20, respectively) in 0.1 M PBS with 0.3% TritonX100. This was followed by incubation with the primary antibodies in a solution with 5% NGS or NRS, 1% bovine serum albumin, (BSA), 0.3% TritonX100 in 0.1 M PBS for one hour at RT, and then at 4 °C overnight with continuous shaking. The following primary antibodies were used (see Table 2): anti-ChAT (Merck, Cat. No. #Ab144p) at a 1:200 dilution and anti-TrkA (Merck, Cat. No. #06-574) at a 1:200 dilution. After washing the sections three times with 0.1 M PBS, they were incubated for 1 h at RT with a secondary antibody: ChAT with rabbit anti-goat IgG antibody, horseradish peroxidase (HRP)-conjugated (Sigma-Aldrich, (St. Louis, MO, USA), Cat. No. AP106P) (5% NGS, 1% BSA, 0.3% TritonX100 in 0.1 MPBS) and TrkA with goat anti-rabbit IgG antibody, HRP-conjugated (Sigma-Aldrich, Cat. No. AP307P) (5% NRS, 1% BSA, 0.3% Triton X100 in 0.1 M PBS) at a dilution of 1:200 for both antibodies. The sections were rinsed three times with 0.1 M PBS and then incubated for 1 h at RT with the Vectastain ABC kit (Vector Laboratories). To visualize primary–secondary antibody complexes, the sections were incubated in PBS containing 3,3′-diaminobenzidine tetrahydrochloride (DAB, Sigma-Aldrich), H_2_O_2_, and NiSO_4_ at a concentration of 0.025%, 0.0125% and 0.04% respectively. The stained sections, representative of the basal forebrain (for details see Section 2.7), were mounted, air-dried and cover-slipped. The controls for the immunohistochemical procedure were obtained by running some slides through the entire procedure with the omission of the primary or secondary antibodies. For all control slides, no specific staining was observed.

### 2.5. Immunohistochemistry for VAChT and Tau in Paraffin Brain Sections

Frozen hemispheres (from 3 to 5 animals per group) were defrosted by moving them from −80 to −20 °C for 6 h, then, for another 6 h, the tissue was stored at 4 °C. After thawing, the brains were immersed into 4% PFA in 0.1 M PBS for 5 days and stored at 4 °C. Tissue was progressed in a series of increasing ethanol concentrations (50% to 100%), processed in pure xylene 3 times, and then embedded in paraffin wax. The brain tissue was sectioned at 6 µm at the desired brain levels (motor cortex and hippocampal CA3) using a rotary microtome (Microm HM325, Leica Biosystems, Nussloch, Germany). For each mouse, eight sections corresponding to the 8 stereotaxic levels of the motor cortex (1.21 to −0.83 anterior/posterior to bregma) and 6 sections corresponding to the 6 stereotaxic levels of the dorsal hippocampus (−1.31 to −2.45 posterior to bregma) were collected on one SuperFrostTM glass slide (Thermo Fisher Scientific, Waltham, MA, USA), respectively. On the first day, the glass slides were deparaffinized (xylene 1, xylene 2, xylene 3, 1:1 xylene: ethanol, 100% ethanol, 100% ethanol, 96% ethanol, 96% ethanol, 70% ethanol, 50% ethanol) and rehydrated in tap water two times for 1 min. Exposure of the antigen was performed in a citric acid solution (pH 6.0) heated to 95–98 °C for 30–35 min, and allowed to cool slowly to RT. The tissue was then washed two times in distilled water and treated with 3% hydrogen peroxide in methanol for 15 min at RT. The sections were washed with water for 10 min and with 0.1 M PBS (pH 7.4) for 5 min. Tissue was incubated at 4 °C overnight with a solution of primary anti-VAChT (1:500, Synaptic Systems Gmbh, Cat. No. 139 103) and anti-tau antibody (1:100, s1D12, Genting TauRx Diagnostic Centre) with 3% bovine serum albumin (BSA) in 0.1 M PBS (Table 2). The next day, after washing three times for 5 min in 0.1 M PBS with 0.3% Triton X 100 (PBST), the tissue was incubated with HRP-conjugated goat anti-rabbit (1:100, Merck, Cat. No. AP307P) for VAChT and goat anti-mouse secondary antibody (1:100, Merck, Cat. No. AP124) or for tau in 5% antibody-appropriate normal rabbit or goat serum (Vector Laboratories, Cat. No. S-5000-20 and S-1000-20, respectively) and 1% BSA in 0.1 M PBST for 1 h at RT. Then, the sections were washed three times for 5 min in 0.1 M PBS. 3,3′-Diaminobenzidine tetrahydrochloride (DAB, Sigma-Aldrich) with H_2_O_2_, and NiSO_4_ at concentrations of 0.025%, 0.0125% and 0.04%, respectively, wasused as chromogen for visualization of the primary–secondary antibody complex. Finally, the samples were mounted using DePeX (Serva).

### 2.6. AChE Histochemistry

To reveal AChE staining in the motor cortex and hippocampus, 40 mm thick free-floating sections (eight sections corresponding to the 8 stereotaxic levels of the motor cortex and six sections corresponding to the 6 stereotaxic levels of the dorsal hippocampus for each of the five mice in the group) were rinsed in double-distilled water and processed in accordance to a modified version of Koelle–Friedenwald method [42,43], i.e., incubated for 1 h (for hippocampus) and 1.5 h (for cortex) at RT on a tissue rocker, in a solution containing ethopropazine, glycine, CuSO_4_·5H_2_O, acetylthiocholine iodide, sodium acetate and glacial acetic acid in double-distilled water. The slides were washed six times for 5 min with water and developed with 1.25% sodium sulfide solution for 1 min. The slides were washed again six times for 5 min with water and incubated with an intensifying solution (1% AgNO_3_, *w*/*v* in water), washed three times for 5 min with water and incubated again two times for 5 min with 5% Na_2_S_2_O_3_ (*w*/*v*), at RT on a tissue rocker. Finally, the slides were rinsed six times for 5 min with water, mounted on slides, immersed in xylene and mounted in DePeX.

### 2.7. Quantitative Analysis of ChAT- and TrkA-Immunoreactive Neurons

The cholinergic basal forebrain regions of interest were pre-defined according to the Mouse Stereotaxic Atlas [41] as the nucleus accumbens (NAc, bregma +1.34 mm AP), vertical limb of the diagonal band of Broca (VDB, bregma +1.18 mm AP), medial septum (MS, bregma +0.98 mm AP), horizontal limb of the diagonal band of Broca (HDB, bregma +0.14 mm AP) and magnocellular basal nucleus (Bregma −0.58 mm AP). Moreover, we also included the striatum, in which the analysis of cholinergic markers was performed on sections approximately 0.14 AP from the Bregma (the level of anterior commissure passage through the midline of the brain). For this final analysis, a slide series from 5 randomly selected animals of each group were chosen. The sections were viewed using a Nikon EclipseNi microscope equipped with an X/Y movement-sensitive stage and Nikon DS-Ri2 video camera attached to a Dell PC. An image analysis NIS Image digitizer and software (Nikon Instruments, Tokyo, Japan) were used for the morphometric analysis of ChAT-ir and TrkA-ir neurons. The boundaries of the structures in the coronal plane were determined microscopically and marked on-line using a computer-aided X–Y plotting system. Subsequently, the counting of the number of neuronal profiles was performed in the digitalized images of each outlined area by two independent and unbiased observers (inter-observer correlation *r* = 0.87). The neurons were counted as cholinergic only if they met the following criteria: neuronal cell bodies were ChAT or TrkA-immunopositive, and the best focused nucleus and at least one dendrite were visible. These criteria were used to exclude any population of non-complete remnants of neurons. All consecutive sections of each series were analyzed. The data are presented as the means ± SD of five animals per group.

### 2.8. Quantitative Analysis of AChE Histochemistry, Anti-VAChT and Anti-Tau Staining

The quantitative analysis of the intensity of staining for AChE, VAChT and tau was performed in the cholinergic projection of the cortex (Bregma 0.86 ± 0.2 mm for the motor cortex) and hippocampus (Bregma −1.94 ± 0.36 mm for the dorsal hippocampus) in accordance with the Mouse Brain Stereotaxic Atlas [41]. AChE, VAChT and anti-tau were analyzed microscopically at 100x magnification using NIS-Elements BR4.30.00 Software (Nikon Instruments, Japan). The measurements were taken from a fixed area of 100,000 ± 1000 mm^2^ in the primary motor cortex and the CA3 hippocampal area, which comprises a cross-section of all the cortical layers and hippocampal stratum lucidum, stratum pyramidale and stratum oriens. The relative protein content was measured as integrated density in 8-bit images using the Fiji image processing package for ImageJ, (NIH National Institutes of Health, Bethesda, MD, USA) [44]. For each individual mouse, data are presented as the mean relative optical intensity (ROI—a semiquantitative densitometric parameter in ImageJ) measured over the two-to-three analyzed sections for each region and antibody (for the WT-HMTM and WT-Combo groups n = 5; for the WT-Veh, L1-Veh and L1-Combo groups n = 4; for the WT-Riva, L1-HMTM and L1-Riva groups n = 3).

### 2.9. Statistical Analysis

All data were analyzed by parametric methods and the values are expressed as the mean ± SD. The morphometric and densitometric data were compared by a factorial two-way analysis of variance (ANOVA) followed by planned comparisons of the selected groups of interest by post hoc multiple-range Newman–Keuls test. There was no a priori assumption of the trends, and all analyses were performed two-tailed. The differences between the groups were considered statistically significant for alpha < 0.05. Statistical analyses were performed using STATISTICA 13.3 software (StatSoft Polska Sp. z o.o., Kraków, Poland). Only significant differences are reported for clarity.

## 3. Results

### 3.1. Effect of Genotype and Treatment on the Morphology of Basal Forebrain Cholinergic Neurons

For greater clarity and contrast, the results of the morphometric analysis of cholinergic neurons are presented for hippocampal (NAc and MS) and cortical circuitries (striatum, VDB, HDB, and NBM). Cholinergic projection neurons originate in the basal forebrain regions such as MS, VDB and HDB or NBM, and send the efferent ones to distal areas such as the hippocampus and cortex. They are dependent on NGF secreted from these regions for their continued survival [45]. There are also cholinergic interneurons in the forebrain. These are located in the striatum and the nucleus accumbens. Consequently, the number of projection neurons was counted in the MS (projecting to the hippocampus) and as the sum of ChAT or TrkA-ir neurons in the VDB, HDB and NBM (projecting to cortical areas), whereas the number of interneurons is presented as the total number of either ChAT- or TrkA-immunoreactive (-ir) neurons in the nucleus accumbens and striatum. Data for the complementary groups are presented in the Appendix A.

The representative microphotographs of the morphology and quantification of the number of ChAT- and TrkA-ir in basal forebrain for the eight core groups are displayed in Figure 2, Figure 3, Figure 4 and Figure 5. WT-Veh mice showed strong cytoplasmic immunoreactivity for ChAT and TrkA in basal forebrain neurons, and their morphology appeared unchanged with HMTM treatment. The cell bodies were densely stained, with tapering varicose dendrites forming an intensely stained neuropil. This was not the case in L1-Veh subjects, where no distinct ChAT and TrkA immunostaining was observed. The remaining ChAT- and TrkA-ir cells appeared shrunken with a markedly reduced intensity of staining and almost invisible dendrites. The administration of HMTM increased the number and morphology of ChAT- and TrkA-ir cells in L1- in comparison to L1-Veh. However, treatment with HMTM did not result in a full restoration of cholinergic neurochemical parameters in the basal forebrain of L1 mice (Figure 2A, Figure 3A, Figure 4A and Figure 5A).

#### 3.1.1. Number of ChAT- and TrkA-ir Neurons in L1 Tau Mice Differ from the Wild-Type Controls

For interneurons in NAc, a factorial two-way ANOVA with the genotype and treatment for the core groups confirmed the highly significant main effect of the genotype (F(1,31) = 18.2; *p* = 0.0002). No other term was reliable (Figure 2B). While there was no treatment effect in each genotype, L1-Veh mice presented with strongly reduced ChAT labeling (t = 2.8, df = 7, *p* = 0.026) relative to the WT.

For interneurons in the striatum (Figure 2C), statistical analysis yielded the main effects of the genotype (F(1,32) = 14.6; *p* = 0.00006) and treatment (F(3,32) = 3.9; *p* = 0.018), but no interaction. Again, the mean number of the ChAT-ir cell bodies was reduced in L1-Veh by 75% (t = 3.7, df = 8, *p* = 0.006 relative to the WT-Veh).

The results for ChAT-ir projection neurons in the MS (Figure 3B) only returned a main effect of genotype (F(1,29) = 12.9; *p* = 0.0012) for the core treatment groups, but no treatment or interaction effects. Again, L1-Veh mice had greatly reduced ChAT labeling (t = 7.9, df = 7; *p* = 0.0001) due to a reduction of >90%. This impression was confirmed in the dataset on VDB/HDB/NBM, with an overall difference between the genotypes (F(3,32) = 12; *p* = 0.0015) due to a strong reduction in ChAT labeling in L1-Veh relative to the WT-Veh (see asterisks in Figure 3C).

Matching results were obtained for TrkA-ir, another specific marker of cholinergic neurons. Again, there was a significant overall genotype effect for both the nucleus accumbens (F(1,32) = 8.6; *p* = 0.006) and striatum (F(1,32) = 5.34; *p* = 0.027), mainly brought about by a significantly lower level of TrkA in L1-Veh (t = 2.3, df = 8; *p* = 0.05 for the nucleus accumbens; t = 3.4; df = 8; *p* = 0.0095 for the striatum). Decreases of about 54% and 66% were observed, respectively (Figure 4B,C).

Similarly, cholinergic projection neurons labeled for TrkA were significantly lower in numbers for L1 cohorts in MS (F1,312) = 6.6, *p* = 0.015) and VDB/HDB/NBM (F(1,32) = 12.8, *p* = 0.001) (Figure 5B,C). For both the hippocampal afferents arising from MS and the cortical input originating in VDB/HDB/NBM, this was mainly due to the significant differences between the WT-Veh and L1-Veh (t = 3.8, *p* = 0.005 for septum; t = 5, df = 8; *p* = 0.001 for VDB/HDB/NBM). These data compellingly confirm the previously reported cholinergic phenotype of L1 mice [9].

Taken together, these data confirm the global loss of basal forebrain acetylcholine-positive neurons in both the hippocampal and cortical projection pathways.

#### 3.1.2. HMTM-15, but Not Rivastigmine-0.5, Recovers Cholinergic Markers in Both Inter- and Projection Neurons in L1 Tau Mice

The administration of HMTM-15 mg but not rivastigmine-0.5 mg recovered the number of ChAT and TrkA-ir neurons in L1 mice (see asterisks in Figure 2, Figure 3, Figure 4 and Figure 5; Appendix A). While HMTM-15 seemed efficacious in the recovery of both ChAT and TrkA in interneurons and projection neurons, riva-0.5 only recovered the TrkA phenotype in MS (Figure 5B). The efficacy of HMTM-15 was such that it recovered cholinergic markers to the WT-Veh levels. While there seemed to be some improvement in cholinergic labeling in L1 after riva-0.5 administration, these data were not reliable due to high variance. Both drugs had no effect on WT mice, apart from riva-0.5 which reduced ChAT labeling in the MS (Figure 3B).

We next asked the question of whether the effects are dose-dependent in L1. A comparison for all L1 treatment conditions is presented in Appendix A. These results confirmed that the two doses of HMTM (5 and 15 mg/kg) recovered the ChAT phenotype in L1 mice in interneurons and projection neurons (Appendix A).

#### 3.1.3. Combination Therapy Prevented the HMTM-15 Benefits in Cholinergic Interneurons and Projection Neurons

Based on the significances in two-way ANOVAs, we compared the beneficial effect of HMTM-15 following the pre-administration of riva-0.5. Statistical comparison between the L1-Veh and L1-combo (riva-0.5 + HMTM-15) showed no improvement in the drug groups for any area scrutinized here (See Figure 2, Figure 3, Figure 4 and Figure 5; Appendix A). While a high data variance precluded significances, the levels of cholinergic markers in L1 combo therapy were always below the WT-Veh animals and often even significantly below L1-HMTM-15 (Figure 5B), indicating an interference between the therapies. Other doses and dose–combinations did not show an effect (Appendix A).

### 3.2. Cholinergic Activity Is Deficient in the Cortex and Hippocampus of L1, and Rescued with HMTM but Not Combination Therapy

We next measured VAChT, the transporter that shuttles reconstituted ACh into secretary organelles ready for synaptic exocytosis upon depolarization of the pre-synaptic bouton. The efficiency of this process is correlated with the strength of the cholinergic projection and its physiological activity [46,47,48]. Projections from the basal forebrain (VDB, HDB, NBM) play an essential role in, for example, motor cortex maturation and control [49,50]. By contrast, cholinergic efferents of the MS are terminated in all regions of the hippocampus, with the strongest synaptic contacts on pyramidal cells in the stratum oriens of CA1 and CA3 [51,52]. Both the cortex and the CA3 field of the hippocampus were selected for analysis (Figure 6A).

As the metabolizing enzyme, acetylcholinesterase (AChE) provides another reliable marker of cholinergic fibers and terminals in the cortex and hippocampus [53,54]. Cortical and hippocampal AChE staining is also coincident with cholinergic terminals that have their origin in the basal forebrain [55,56,57,58], and vary in their laminar patterning [59,60]. In the hippocampus, AChE was most intensely stained in the stratum pyramidale and with a lower intensity in the adjacent stratum radiatum and stratum oriens (Figure 7A).

#### 3.2.1. The expressions of VAChT and AChE in L1 Tauopathic Mice Differ from the Wild-Type Controls

In the control WT-Veh mice, cortical staining tended to coalesce into bilaminar patterns of intensive staining that was observed in cortical layers II–III and V, while the other layers were only weakly stained. In the hippocampal formation, VAChT-immunoreactive fibers were observed predominantly in the CA3 region. The strong immunoreactivity of VAChT in nerve endings and fibers clearly identified the afferent terminals of cholinergic axons on pyramidal cell somas in the stratum pyramidale, but also on the dendrites of pyramidal cells in the stratum oriens and stratum radiatum of CA3 (Figure 6A). This characteristic pattern of VAChT immunostaining was disrupted in L1 mice, with a decreased intensity of VAChT-ir observed in both the cortex and hippocampus. VAChT-ir was restored through the administration of HMTM in L1, most prominently in CA3 (Figure 6A).

AChE staining in the motor cortex returned a bilaminar pattern of moderately intense AChE staining in cortical layers II–III and V in the WT-Veh controls. Contrastingly, an aberrant labeling pattern was observed in L1-Veh mice, in which the cortical staining intensity was generally reduced (Figure 7A). Similar results were obtained for the hippocampus, where the AChE staining density was strongly reduced in L1-Veh mice relative to the WT-Veh. The administration of HMTM caused an increase in the enzyme activity to a level comparable with the WT-Veh, both in the cortex and hippocampus (Figure 7A).

Quantitative densitometry was performed on both of these markers. The relative optical intensity (ROI) of the VAChT-ir (Figure 6B) showed differences that corresponded with the reduced cholinergic cell number of the basal forebrain reported above. For the core groups, two-way ANOVAs revealed a significant difference between the groups for the cortex (F(7,32) = 68.3, *p* < 0.0001) and the hippocampus (F(7,32) = 24.7, *p* < 0.0001), but also the main effects of treatment and interactions (cortex—treatment: F(3,32) = 4.3; *p* < 0.0046—interaction: F(3,32) = 4.7; *p* = 0.01; hippocampus—treatment: F(3,32) = 12.3; *p* < 0.0001—interaction: F(3,32) = 8; *p* = 0.0004). The differences in the cortical VAChT-ir phenotype between the WT and the L1 vehicle groups were confirmed by a two-tailed planned comparison (t = 4.2, df8; *p* = 0.003; Figure 6B). For the hippocampus, the VAChT-ir phenotype was also confirmed for the L1-Veh groups relative to the WT-Veh (t = 5.2, df = 7; *p* = 0.0013; Figure 6C).

Similar results were found for AChE labeling (Figure 7A), with L1 subjects showing vastly diminished AChE positivity compared with the WT in both the cortex (Figure 7B) and the hippocampus (Figure 7C). As highlighted for VAChT, there were significant main effects of the genotype for both the cortex (F(1,32) = 9; *p* = 0.005) and the hippocampus (F(1,32) = 12; *p* = 0.002), but the treatment and interactions were reliable only for the former (treatment: F(3,32) = 7.85; *p* = 0.0005, interaction: F(3,32) = 5.2; *p* = 0.005). Two-tailed planned comparisons confirmed the significantly reduced AChE levels in both the cortex (t = 3.3, df = 8; *p* = 0.01) and the hippocampus (t = 5.2, df = 8; *p* = 0.0008).

Overall, these data confirm our previously reported cholinergic deficit in L1 mice [9] in a new and independent cohort.

#### 3.2.2. HMTM, but Not Rivastigmine, Reverses Impaired VAChT and AChE Expression in L1 Tau Mice

Long-term treatment with HMTM-15 or rivastigmine-0.5 produced negligible changes in VAChT labeling in the motor cortex of L1 (Figure 6A; levels did not differ from the L1-Veh). Intriguingly, VAChT labeling intensity was enhanced in the cortex of WT cohorts exposed to rivastigmine-0.5 by 38% relative to the WT-Veh (t = 6.3, df = 8; *p* = 0.0002; Figure 6B). No such effects were observed for the hippocampus (Figure 6C). In the hippocampus, however, there was a strong recovery of the ROI for VAChT in L1-HMTM-15 subjects (92% increase, t = 6, df = 8; *p* = 0.0001), but only a 42% increase in the L1-rivastigmine-0.5 cohort (t = 5, df = 8; *p* = 0.001; see asterisks in Figure 6C). Nevertheless, this effect seemed to be dose-independent and was also observed when drugs were combined (see Appendix A).

The AChE staining density remained unchanged following HMTM-15 treatment in WT-Veh subjects in both the cortex and hippocampus (Figure 7B,C). L1 mice responded very well to HMTM-15 by increasing the levels of AChE in the cortex by 50% (t = 3.5, df = 8; *p* = 0.008) and by 43% in the hippocampus (t = 5.1, df = 8; *p* = 0.0009). These results indicate a clear normalization of AChE labeling to WT levels. As would be expected from an AChE inhibitor, rivastigmine-0.5 lowered the staining density for AChE in both the cortex (t = 3, df = 8; *p* = 0.02) and hippocampus (t = 6.5, df = 8; *p* = 0.0002) in WT subjects (Figure 7B,C). In contrast, no change occurred in L1 mice treated with rivastigmine-0.5 and the AChE levels remained at the reduced level typical for L1 tau mice, even when different doses were considered (see Appendix A).

#### 3.2.3. Rivastigmine Interferes Negatively with the Beneficial Effect of HMTM on VAChT and AChE Expression in L1 Tau Mice

When rivastigmine-0.5 was administered first for 5 weeks and then combined with HMTM-15 for another 6 weeks, the typical beneficial effects observed for HMTM-15 as a single therapy were not seen. This was particularly obvious for VAChT (Figure 6C) and AChE (Figure 7C) in the hippocampus, where HMTM-15 strongly reversed the deficits (see above), but combination therapy at these doses failed. In the hippocampus, this was mainly due to rivastigmine at 0.5 mg/kg, which, when given alone or in combination, did not reverse the AChE phenotype in L1 (t’s < 1.7; see Appendix A). All other drug doses and drug combinations, however, were effective (all t’s > 2.5). Similar results were obtained for the cortex (Figure 6B and Figure 7B), but HMTM-15 was not as efficient as for hippocampus. Nevertheless, the combination with rivastigmine-0.5 clearly prevented any HMTM-15 improvement. Smaller doses were not effective (Appendix A).

A different result was observed for WT cohorts, where all the effects of HMTM-15 were maintained in the presence of rivastigmine-0.5. This provides compelling evidence for an interference of rivastigmine in the efficacy of HMTM, at least at the concentrations administered here.

### 3.3. HMTM Affects the Level of Tau in L1 Mice

We previously highlighted the fact that the administration of MTC or HMTM can reduce tau loading in L1 mice [30]. We therefore sought to confirm this by using the newly developed antibody S1D12, which recognizes an epitope within tau337–355. In the motor cortex and the hippocampus, the intensity of immunostaining with S1D12 was distinctly higher in L1-Veh mice than in the WT-Veh group (Figure 8A and Figure 9A). Both the WT controls and L1 tau mice expressed S1D12-anti-tau immunoreactivity mainly in fibers and processes including axons. However, WT-Veh mice presented with long projections penetrating the cortical layers (Figure 8B), while in L1-Veh, these processes were short with scattered segments (Figure 8B). We take this as evidence for axonal dystrophy in L1 mice. In addition, L1 mice showed dark granular deposits (puncta) in close proximity of pyramidal cells, both in the cortex and the hippocampus, which may indicate an early accumulation of tau in the synapses of projection cells (Figure 8B and Figure 9B).

#### 3.3.1. Cortical Tau Immunoreactivity in L1 Tau Mice Is Normalized by HMTM-15 and Rivastigmine-0.5, but Not by Combination Therapy

The quantitative analysis confirmed the macroscopic observations. For the cortex, a two-way ANOVA of the core groups revealed a significant difference between treatments and an interaction with the genotype (F’s > 13, *p* < 0.0001), which was confirmed by a planned post hoc analysis to yield a reliably higher level of tau labeling in L1-Veh relative to the WT-Veh (t = 3.9, df = 6; *p* = 0.007). While this was reversed in L1 by 44% in the HMTM-15 cohort (t = 3.8, df = 6; *p* = 0.009), as predicted [30], there was only a trend for the lower dose of HMTM 5 mg/kg (*p* = 0.09; Appendix A). We also found a small reduction in tau for rivastigmine-0.5 (t = 3.5, df = 5; *p* = 0.02) but not at lower doses. Nevertheless, the pre-administration of rivastigmine-0.5 followed by HMTM-15 prevented this normalization (t < 1), suggesting a negative interaction between the two compounds (see asterisks in Figure 8C). This was corroborated by our complementary groups in that the block of HMTM efficacy by rivastigmine was seen for all drug dose combinations (see Appendix A). The only exception was the low doses of 0.1 mg/kg rivastigmine with 5 mg/kg HMTM which also normalized tau levels in the cortex (t = 4.3, df = 6; *p* = 0.005).

Another unpredicted observation was the over 100% increase in tau labeling in the WT animals treated with HMTM-15. The labeled tau in the WT is likely to be qualitatively different from that in L1 in that it consists of murine tau only, and was significantly heightened relative to the WT-Veh (t = 6.1, df = 7; *p* = 0.0005).

#### 3.3.2. Hippocampal Tau Immunoreactivity in L1 Tau Mice Is Normalized by HMTM-15, but Not by Rivastigmine-0.5 or Combination Therapy

For the hippocampus, the results were comparable to those observed in the cortex. A two-way ANOVA for the core groups showed the significant main effects of the genotype, treatment and an interaction between these factors (F’s < 3.9; p’s < 0.0026) (Figure 9C). In the vehicle groups, a significantly enhanced S1D12 labeling occurred for L1 (t = 4.4, df = 6; *p* = 0.0005), which was reduced by 66% in the presence of HMTM-15 (t = 5.8, df = 6; *p* = 0.001), and also by the lower dose of HMTM-5 (t = 4.7, df = 6; *p* = 0.004; Appendix A), but not when combined with rivastigmine-0.5 for any dose combination (Appendix A). Again, rivastigmine-0.5 alone also exerted some beneficial effects and reduced tau labeling in L1 subjects, albeit not significantly (see asterisks in Figure 9C).

As revealed in the cortex, HMTM-15 also increased the levels of murine tau (t = 3, df = 7; *p* = 0.02) by 56% in the hippocampus of the NMRI vehicle group. These data stress the prevention efficacy of HMTM when rivastigmine is pre-administered to tau mice.

## 4. Discussion

The broader goals of the current studies were to(1) provide further in vivo evidence of an interaction between tauopathy and cholinergic atrophy in tau transgenic L1 mice, (2) test whether HMTM administration attenuates the development of tauopathy and worsening of the cholinergic status in an animal AD model and (3) provide a better insight into the decreased therapeutic efficacy when HMTM is administered against the background of a symptomatic rivastigmine treatment.

We report three main findings: (i) tau transgenic mice develop an impairment of the cholinergic system and this confirms our previous findings [9,15]; (ii) HMTM treatment partially reversed this impairment and lessened the tauopathy in L1; (iii) combined HMTM and rivastigmine treatment is less efficacious than HMTM monotherapy, with the cholinesterase inhibitor completely blocking any beneficial effects of HMTM in some instances. Collectively, these data suggest that, in L1 tau mice with a severe expression of human tau [36] and a cholinergic deficit [9], the loss of the cholinergic phenotype in basal forebrain neurons is probably reversible through the administration of the tau aggregation inhibitor HMTM and, to a lesser extent, the symptomatic drug rivastigmine. Secondly, we confirmed a negative interaction between the two treatments, i.e., the administration of HMTM against prior long-term treatment with rivastigmine [15,34], in agreement with clinical observations [5,33,61,62].

### 4.1. Tau Transgenic L1 Mice as a Model for Early Stages of Alzheimer’s Disease

The basal forebrain cholinergic system occupies a central role in normal cognition and age-related cognitive decline, including dementias such as AD. The cholinergic hypothesis of AD is centered on the progressive loss of limbic and neocortical cholinergic innervation [63], and is supported by neurotrophic deficiencies given the dysregulations of nerve growth factor signaling [64]. The progressive and selective degradation of the basal forebrain cholinergic system has been well documented during the onset of AD in humans and animal models, both in vivo and in vitro [65,66,67], and neurofibrillary degeneration is believed to be the primary cause of the loss of cholinergic innervation [68,69,70,71] and a concomitant loss of cognitive function characteristic of AD [13,72]. Abnormally phosphorylated tau in the form of oligomers or early neurofibrillary tangles in the basal forebrain contributes to both the initiation and/or progression of neurodegeneration in AD [73,74]. Basal forebrain cholinergic neurons are among the earliest to show tau oligomers [75,76] and present with the dysregulation of genes encoding neurotrophic and neurotransmitter signaling proteins in the AD brain [77].

The findings reported here confirm that the integrity of the basal forebrain cholinergic system is deteriorated in tau transgenic L1 mice. This was indicated by the impaired cholinergic neuronal morphology, a reduced number of ChAT positive cells [9] and a progressive impairment of spatial learning when aged three-to-six months [30]. Using a number of additional cholinergic markers, we expanded this reduced functioning not only based on a significant decrease in the number of ChAT- and TrkA-stained neurons in the basal forebrain, but also on the lowered density of VAChT immunolabeling and histological staining of AChE of cortical and hippocampal efferents. The reduction in morphometric parameters of ChAT and TrkA immunolabeling was particularly severe in large projection neurons located in the MS, VDB, HDB and NBM, as well as the number of small cholinergic interneurons present in the NAc and ST of L1 mice. These data provide unambiguous evidence for the lowering of cholinergic function in the basal forebrain of L1 mice.

The target projection zones (motor cortex and hippocampal CA3) were also explored, with labeling for (i) VAChT, which mediates the transfer of ACh from the cytoplasm into secretory synaptic vesicles; and (ii) AChE, a type B carboxylesterase that rapidly hydrolyzes ACh in brain cholinergic synapses and terminates neuronal signaling of extracellular ACh. Both markers were lowered in L1-Veh compared to wild-type mice in the cortex and CA3. These results provide compelling evidence that the disturbances in the cholinergic system of L1 mice concern not only structural changes in cholinergic neurons of the basal forebrain, but also functional changes at their projection targets, and hence serve as a good model for AD. These traits coincided with enhanced tau immunoreactivity in L1 mice, suggesting that the underlying tau pathology is driving this cholinergic loss.

### 4.2. The Therapeutic Effect of HMTM Is Similar in Cholinergic Projections to the Hippocampus and Neocortex: Blocking by Rivastigmine

The reduction in the morphometric parameters of ChAT- and TrkA-immunolabeled neurons was partially or completely reversed by the prolonged HMTM administration as a monotherapy. This indicates that for untreated L1 tau mice, the basal forebrain cholinergic neurons have a lowered expression of ChAT and TrkA receptors and that, following HMTM treatment, they recover their ChAT and TrkA expression to detectable levels. The administration of HMTM to L1 mice also increased the levels of VAChT and AChE, in agreement with the complete rescue of the cholinergic system with a significant cholinergic tone in the cortex and hippocampus [15]. At the same time, HMTM also attenuated the tau load in both pathways, further corroborating the notion that the driver for the pathology of the cholinergic system is the formation of tau aggregates.

The beneficial effect of HMTM was less evident when combination therapy was used. In the present study, the effects of HMTM on cholinergic function were inhibited by pre-treatment followed by concurrent treatment with rivastigmine, a prototypic AChE inhibitor, and HMTM. The administration of HMTM in combination with rivastigmine was also less effective in reducing tau protein levels in L1 mice, although monotherapy with HMTM significantly reduced tauopathy symptoms present in untreated L1 mice. This reduction in the pharmacological efficacy of HMTM, when given as an add-on to symptomatic treatment in humans, was observed in several other studies using the L1 tau model [15,34] and in clinical trials [5,61]. However, similar studies using other AD treatments are widely missing, and combination therapies have concentrated on symptomatic treatments but the benefits of combinations with AChE inhibitors or memantine remain controversial [78,79]. In clinical trials, participants were frequently being prescribed one of the symptomatic treatments currently approved for Alzheimer’s disease (donepezil, rivastigmine, memantine); interference with HMTM might be expected in experimental models with each of these drugs, but further work is required to understand the mechanism of action for each drug. Several results suggest that rivastigmine, donepezil and galantamine differ in their pharmacology. While donepezil and galantamine do not inhibit BuChE and are associated with an increase in CSF AChE protein levels, rivastigmine provides a sustained inhibition of both AChE and BuChE [80]. However, we have no evidence that this difference in the mechanisms of action of cholinesterase inhibitors is important in modulating the efficacy of add-on therapy with HMTM.

To our knowledge, we are the first to investigate Riva/HMTM combination treatment in tau models of AD at multiple levels/systems of the brain, and have demonstrated decreased pharmacological efficacy of combination therapy in the regulation of hippocampal acetylcholine [15], synaptosomal glutamate release and mitochondrial activity [34,81], and the presynaptic expression of SNARE proteins involved in synaptic vesicle docking and fusion [82]. Although a decrease in brain MT^+^ was observed in animals undergoing long-term rivastigmine treatment [80], the MT levels were still within the therapeutic range and should have been effective. The block of HMTM activity is therefore not due to the decreased bioavailability of HMTM/MT^+^, but appears due to neuronal interactions. We have previously postulated a homeostatic adjustment of the nervous system to the long-term availability of rivastigmine/acetylcholinesterase inhibitors in general [34], and these data provide further support for such a mechanism.

### 4.3. Cholinergic System as a Potential Disease-Modifying Target for HMTM

The central question arising from this research is how does the cholinergic phenotype recover with HMTM? Here, we offer two alternative scenarios. (1) It is conceivable that our treatment with HMTM was initiated at the very timepoint when atrophy of cholinergic neurons began in L1 mice, and HMTM protected these cells from degeneration. Our data suggest that neurodegeneration in L1 is greatly dependent on the build-up of tau in cholinergic neurons and could indeed have been prevented by the lowering of tau with this therapy. This is consistent with the failure of rivastigmine alone or in combination with HMTM to fully recover/protect the cholinergic phenotype that emerged in L1 mice, despite therapeutically effective doses being administered. Evidence comes from our AChE measurements in WT mice which were strongly decreased by rivastigmine in both the cortex and hippocampus at a 0.5 but not 0.1 mg/kg dose. (2) An opposite mechanism would be that the HMTM treatment restored the cholinergic phenotype that was lost in the L1 mice. This explanation is supported by our previous observation that the treatment to replenish NGF recovered acetylcholine in aging rats to the level of young adult subjects [82]. Thus, the neuroprotective effects of HMTM are akin to the substances that provide continuous trophic support for the cholinergic system [83,84,85]. Whether HMTM exerts direct or indirect trophic support remains elusive. Consequently, impairment of the cholinergic system in L1 appears to be due to the lowered functional expression of choline acetyltransferase and tyrosine kinase receptors rather than the degeneration of neuronal cell bodies. And it is this functional recovery which is brought about by both doses (5 and 15 mg/kg) of HMTM in both inhibitory and projection neurons, but not by rivastigmine or combination treatment. Here, we provide strong and supporting evidence for the notion that HMTM is mechanistically different and not acting via the inhibition of acetylcholinesterase [15] (Figure 7). Rather, HMTM appears to increase AChE activity. How this rescue of cholinergic cells from a quasi-silent /degenerative state is mechanistically achieved by HMTM requires a more in-depth study. However, it suggests that our L1 model truthfully mimics the cholinergic deficit that is a principle and early feature of AD and, in part, due to the suppression of cholinergic markers in the absence of cell death [69]. Since similar observations have recently been reported for AD [86], our results provide a functional explanation as to why HMTM monotherapy induced an arrest of cognitive decline and brain atrophy in clinical AD [5,6]. Such a rescue of cholinergic cells and lowering of tau levels are two potential mechanisms by which HMTM may halt the progression of AD [34,80,81]. It would similarly explain why the upregulated cholinergic tone with rivastigmine pretreatment negatively interferes with these mechanisms.

## 5. Conclusions

This research was motivated by clinical observations of the tau aggregation inhibitor HMTM being effective as a mono but not add-on therapy on the background of symptomatic treatment [5,61]. We replicated this here in an experimental model, providing compelling evidence that monotherapy with different doses of HMTM rescues the cholinergic phenotype in both projection and inhibitory circuits, but that this effect was prevented by the pre-administration of rivastigmine over several weeks. This negative interaction may be explained by the hyperactivation of the cholinergic tone which would interfere with the trophic rescue of ACh neurons by HMTM. In addition, this trophic support and recovery of basal forebrain neurons to normal cholinergic function would explain the arrest of both the cognitive decline and brain atrophy observed in clinical trials [87].

## Figures and Tables

**Figure 1 cells-13-00642-f001:**
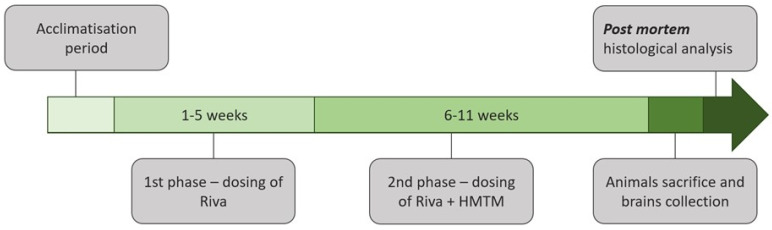
Schematic representation of the study design. A detailed description of the research protocol is provided in the Section 2 and Table 1. HMTM—hydroxymethylthionine mesylate; Riva—rivastigmine.

**Figure 2 cells-13-00642-f002:**
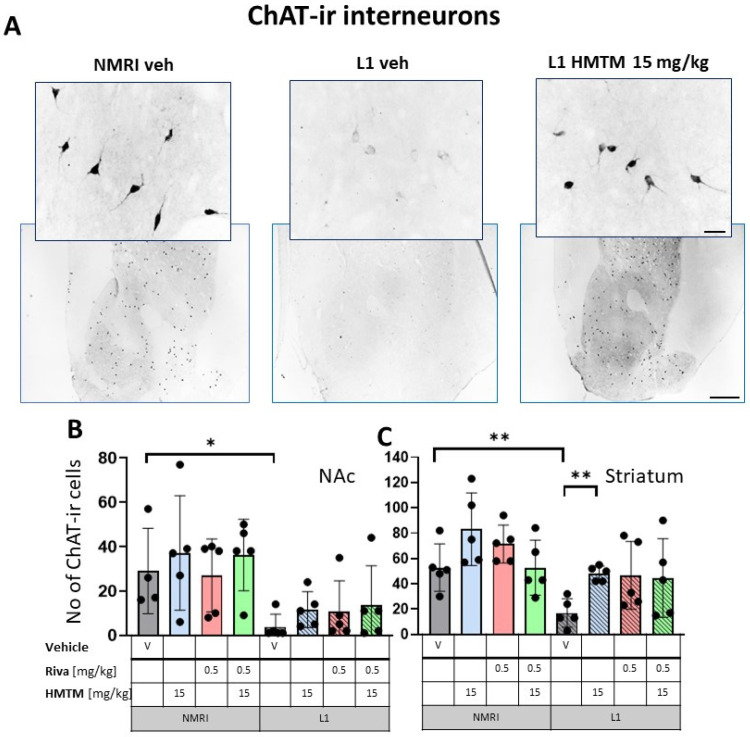
Choline acetyltransferase (ChAT) immunohistochemistry in the basal forebrain interneurons of wild-type and L1 mice. The core groups were treated as indicated with the vehicle, HMTM, rivastigmine or a combination of the two compounds. (**A**) Representative images of the ChAT immunohistochemical staining of interneurons. Sums of ChAT-ir interneurons counted in (**B**) the nucleus accumbens and (**C**) striatum. Values are presented as the mean ± SD with scatter plots of individual data. (**B**) For interneurons in the nucleus accumbens, a factorial two-way ANOVA confirmed a significant main effect of the genotype (F(1,31) = 18.2; *p* = 0.0002). No other term was reliable. L1-Veh mice presented with strongly reduced ChAT labeling (t = 2.8, df = 7, *p* = 0.026) relative to WT. (**C**) For interneurons in the striatum, statistical analysis yielded the main effects of the genotype (F(1,32) = 14.6; *p* = 0.00006) and treatment (F(3,32) = 3.9; *p* = 0.018), but no interaction. The mean number of the ChAT-ir neurons was reduced in L1-Veh by 75% (t = 3.7, df = 8, *p* = 0.006 relative to WT-Veh). While there was a substantial loss of ChAT labeling in L1 mice relative to NMRI, this phenotype has been recovered when HMTM is administered. The combination of rivastigmine and HMTM did not lead to added benefits. Statistical significance between the groups was calculated using the Newman–Keuls test (* *p* < 0.05, ** *p* < 0.01). Scale bars: 100 μm for the basic microphotographs and 50 μm for the insertions; n = 5 for all groups.

**Figure 3 cells-13-00642-f003:**
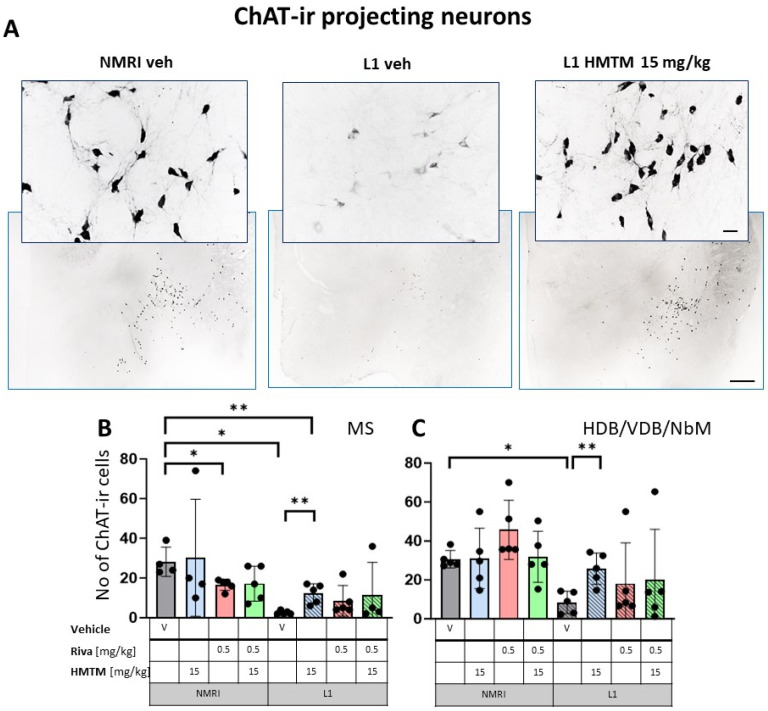
Choline acetyltransferase (ChAT) immunohistochemistry in the basal forebrain projecting neurons of wild-type and L1 mice. The core groups were treated as indicated with the vehicle, HMTM, rivastigmine or a combination of the two compounds. (**A**) Representative images of the ChAT immunohistochemical staining of projecting neurons. Sums of ChAT-ir projection neurons counted in (**B**) the medial septum and (**C**) VDB, HDB and the substantia innominata-magnocellular basal nucleus (NBM). Values are presented as the mean ± SD with scatter plots of individual data. (**B**) Results for ChAT-ir projection neurons in the medial septum demonstrated the main effect of the genotype (F(1,29) = 12.9; *p* = 0.0012) for the core treatment groups, but no treatment or interaction effects. L1-Veh mice had greatly reduced ChAT labeling (t = 7.9, df = 7; *p* = 0.0001) due to a reduction of >90%. (**C**) For cholinergic neurons (ChAT-ir) in VDB/HDB/NBM, the analysis showed a difference between genotypes (F(3,32) = 12; *p* = 0.0015) due to a strong reduction in ChAT labeling in L1-Veh relative to WT-Veh. While there is a dramatic loss of ChAT labeling in L1 mice relative to NMRI, this phenotype was recovered when HMTM was administered. The combination of rivastigmine and HMTM did not lead to added benefits. Statistical significance between the groups was calculated using the Newman–Keuls test (* *p* < 0.05, ** *p* < 0.01). Scale bars: 100 μm for the basic microphotographs and 50 μm for the insertions; n = 5 for all groups.

**Figure 4 cells-13-00642-f004:**
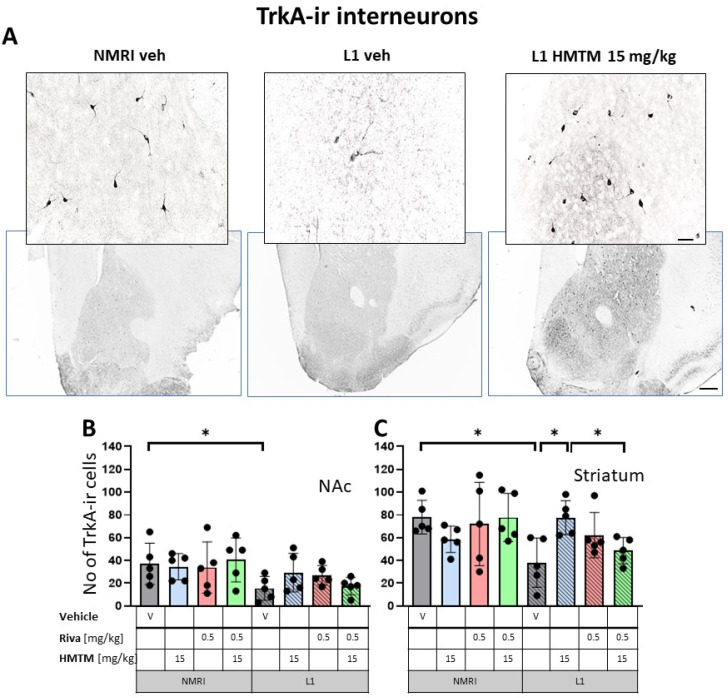
High-affinity nerve growth factor receptor (TrkA) immunohistochemistry in the basal forebrain interneurons of wild-type and L1 mice.. The core groups were treated as indicated with the vehicle, HMTM, rivastigmine or a combination of the two compounds. (**A**) Representative images of the immunohistochemical staining against TrkA for interneurons. Sums of TrkA-ir interneurons counted in (**B**) the nucleus accumbens and (**C**) striatum. Values presented as the mean ± SD with scatter plots of individual data. There was a significant overall genotype effect for both (**B**) the nucleus accumbens (F(1,32) = 8.6; *p* = 0.006) and (**C**) striatum (F(1,32) = 5.34; *p* = 0.027) mainly brought about by a significantly lower level of TrkA in L1-Veh (t = 2.3, df = 8; *p* = 0.05 for the nucleus accumbens; t = 3.4; df = 8; *p* = 0.0095 for the striatum). (**C**) HMTM administration protects TrKA-ir neurons in the striatum of L1 mice while rivastigmine and combined treatment did not affect the mean number of neurons. Statistical significance between the groups was calculated using the Newman–Keuls test (* *p* < 0.05). Scale bars: 100 μm for the basic microphotographs and 50 μm for the insertions. n = 5 for all groups.

**Figure 5 cells-13-00642-f005:**
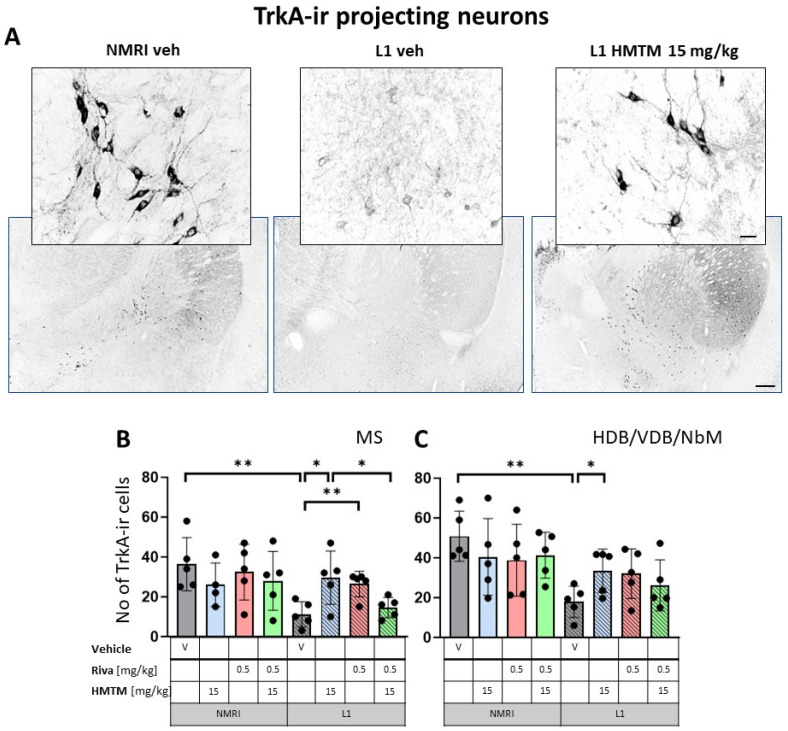
High-affinity nerve growth factor receptor (TrkA) immunohistochemistry in the basal forebrain projecting neurons of wild-type and L1 mice. The core groups were treated as indicated with the vehicle, HMTM, rivastigmine or a combination of the two compounds. (**A**) Representative images of TrkA immunohistochemical staining of projecting neurons in the substantia innominata-magnocellular basal nucleus (NBM). Sums of ChAT-ir projection neurons counted in (**B**) the medial septum, and (**C**) VDB, HDB and NBM. Cholinergic projection neurons labeled for TrkA showed a significant decrease in numbers in (**B**) the medial septum (F1,31 = 6.6, *p* = 0.015) and (**C**) VDB/HDB/NBM (F(1,32) = 12.8, *p* = 0.001). In both cases, the reduction was primarily attributed to significant differences between WT-Veh and L1-Veh (t = 3.8, *p* = 0.005 for medial septum; t = 5, df = 8; *p* = 0.001 for VDB/HDB/NBM). HMTM treatment significantly increased the number of TrkA-ir neurons in both the medial septum and VDB/HDB/NBM in L1 mice. The same effect was observed in L1 groups treated with rivastigmine, but only in the medial septum. The combination of rivastigmine and HMTM did not lead to added benefits. Statistical significance between the groups was calculated using the Newman–Keuls test (* *p* < 0.05, ** *p* < 0.01). Values are presented as the mean ± SD with scatter plots for individual data. Scale bars: 100 μm for the basic microphotographs and 50 μm for the insertions. n = 5 for all groups.

**Figure 6 cells-13-00642-f006:**
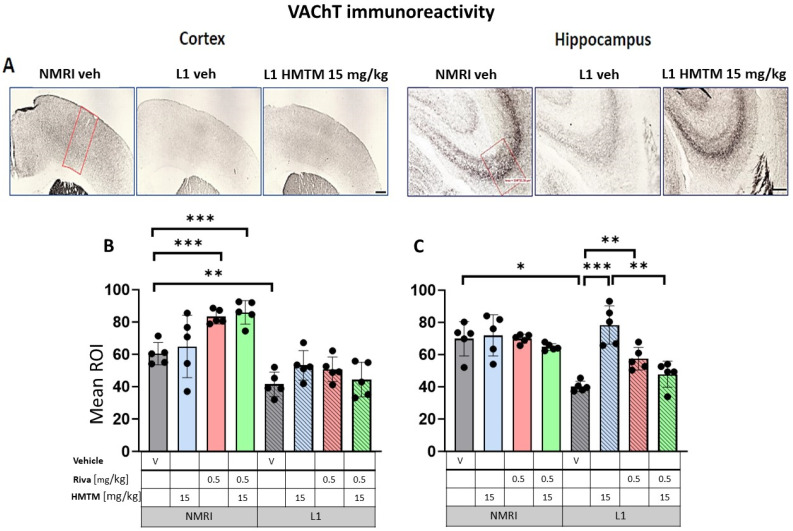
Vesicular acetylcholine transporter (VAChT) immunohistochemistry in the cortex and hippocampus of wild-type and L1 mice. The core groups were treated as indicated with the vehicle, HMTM, rivastigmine or a combination of the two compounds. (**A**) Representative images of the VAChT immunohistochemical staining in the motor cortex (**left panel**) and the CA3 field of hippocampus (**right panel**). Mean relative optical density (mean ROI) of the VAChT staining in (**B**) the motor cortex and (**C**) hippocampal CA3. Two-way ANOVAs indicated a significant difference between the groups in (**B**) the cortex (F(7,31) = 63.5, *p* < 0.0001) and (**C**) hippocampus (F(7,30) = 20.8, *p* < 0.0001), with the main effects of treatment and interactions observed (cortex—treatment: F(3,31) = 4.3; *p* < 0.013, interaction: F(3,31) = 4.7; *p* = 0.008; hippocampus—treatment: F(3,30) = 11.25; *p* < 0.0001, interaction: F(3,30) = 7.6; *p* = 0.0007). A two-tailed planned comparison confirmed the cortex phenotype between the vehicle groups of WT and L1 (t = 3.8, df = 7; *p* = 0.007). Additionally, for the hippocampus, the VAChT-ir phenotype was confirmed for the L1-Veh groups relative to the WT-Veh (t = 5.2, df = 7; *p* = 0.0013). (**B**) No recovery of VAChT-positive neurons was observed in the cortex of L1 across all HMTM treatment groups. However, the control groups treated with rivastigmine or with a combination of rivastigmine and HMTM had elevated levels of the mean ROI. (**C**) For the hippocampus, there was a clear recovery of VAChT-ir in the HMTM L1 group (t = 6, df = 7; *p* = 0.0005). The same effect was observed in L1 groups treated with rivastigmine (t = 4.3, df = 7; *p* = 0.003) and a combination of both compounds, but this effect was weaker than for HMTM. Treatments did not affect levels in the control mice. Statistical significance between the groups was calculated using the Newman–Keuls test (* *p* < 0.05, ** *p* < 0.01, *** *p* < 0.001). Values are presented as the mean ± SD with scatter plots for individual data. Scale bars: 200 µm for cortex microphotographs and 100 µm for hippocampus microphotographs. For the WT-Veh and WT-Combo groups n = 5; for the L1-Veh and L1-Combo groups n = 4; for the WT-Riva, L1-LMTM and L1-Riva groups n = 3.

**Figure 7 cells-13-00642-f007:**
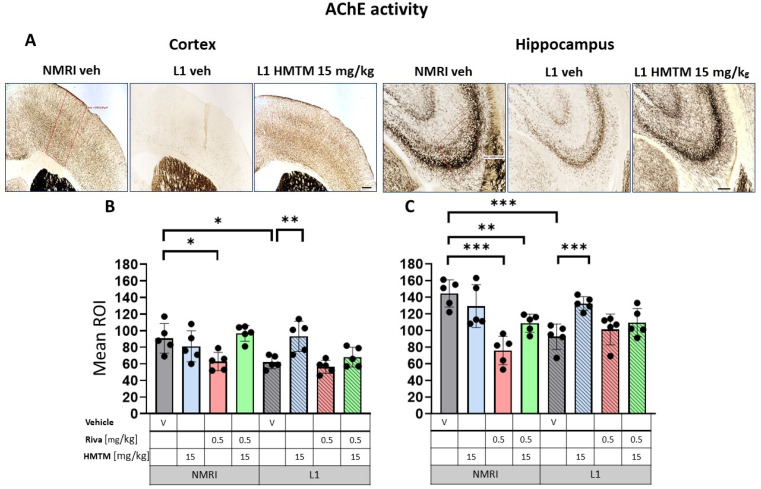
Acetylcholinesterase (AChE) activity in the cortex and hippocampus of wild-type and L1 mice. The core groups were treated as indicated with the vehicle, HMTM, rivastigmine or a combination of the two compounds. (**A**) Representative images of AChE activity in the motor cortex (**left panel**) and for the CA3 of the hippocampus (**right panel**). The mean relative optical density (mean ROI) of AChE staining in the motor cortex (**B**) and hippocampal CA3 (**C**). For the AChE, there were noticeable effects based on genetic differences in both (**B**) the cortex (F(1,32); *p* = 0.005) and (**C**) hippocampus (F(1,32) = 12; *p* = 0.002). However, the treatment and interactions were significant only for the cortex (treatment: F(3,32) = 7.85; *p* = 0.0005, interaction: F(3,32) = 5.2; *p* = 0.005). Two-tailed planned comparisons confirmed a significant reduction in AChE levels in both the cortex (t = 3.3, df = 8; *p* = 0.01) and hippocampus (t = 5.2, df = 8; *p* = 0.0008). In both structures, there was a significant recovery of AChE levels in the L1 groups treated with HMTM (cortex—t = 3.5, df = 8; *p* = 0.008; hippocampus—t = 5.1, df = 8; *p* = 0.0009). The combination of rivastigmine and HMTM did not lead to added benefits in L1 groups. Treatment with rivastigmine decreased the intensity of AChE staining in the WT in both the cortex (t = 3, df = 8; *p* = 0.02) and hippocampus (t = 6.5, df = 8; *p* = 0.0002) but not in L1. Statistical significance between the groups was calculated using the Newman–Keuls test (* *p* < 0.05, ** *p* < 0.01, *** *p* < 0.001). Values are presented as the mean ± SD with scatter plots for individual data. Scale bars: 200 µm for cortex microphotographs and 100 µm for hippocampus microphotographs. n = 5 for all groups.

**Figure 8 cells-13-00642-f008:**
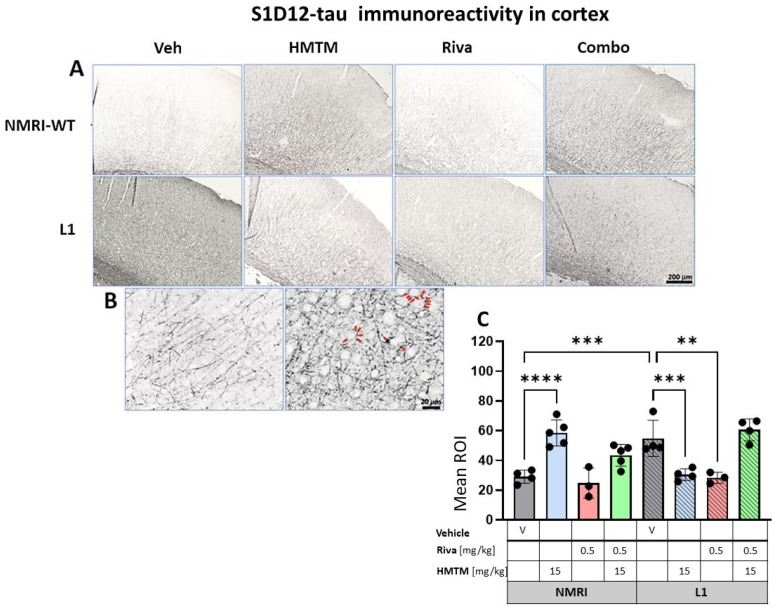
S1D12-tau immunohistochemistry in the motor cortex of wild-type and L1 mice. The core groups were treated as indicated with the vehicle, HMTM, rivastigmine and a combination of the two compounds. (**A**) Representative images of S1D12 immunoreactivity in the cortex. (**B**) High magnification of anti-tau cortical staining in vehicle-treated NMRI-WT (on the **left**) and L1 (on the **right**) mice showing a different pattern of tau labeling in neural processes in both groups. In L1 mice (right image), the presence of dark granular deposits (some examples indicated by arrows) in the vicinity of pyramidal cells surface suggests elevated tau levels at the synapses. (**C**) The mean relative optical density (mean ROI) of S1D12 staining in the cortex. An ANOVA of the core groups revealed a significant difference between treatments and an interactions with the genotype (F’s > 13, *p* < 0.0001). Two-tailed planned comparisons confirmed a significant increase in tau levels in the cortex of L1 mice compared to NMRI (t = 3.9, df = 6; *p* = 0.007), and this phenotype was recovered upon administration of HMTM in L1 animals (t = 3.8, df = 6; *p* = 0.009). The same effect, albeit weaker, was observed in L1 mice treated with rivastigmine (t = 3.5, df = 5; *p* = 0.02). The combination of rivastigmine and HMTM did not lead to added benefits (t < 1). NMRI-WT treated with HMTM had a significant increase in tau intensity in comparison with the NMRI vehicle group (t = 6.1, df = 7; *p* = 0.0005). Values are presented as the mean ± SD with scatter plots for individual data points. Statistical significance between the groups was calculated using the Newman–Keuls test (** *p* < 0.01, *** *p* < 0.001, **** *p* < 0.0001). Scale bars: 200 µm for cortex microphotographs and 20 μm for the insertions. For the WT-Veh and WT-Combo groups n = 5; for the L1-Veh and L1-Combo groups n = 4; for the WT-Riva, L1-LMTM and L1-Riva groups n = 3.

**Figure 9 cells-13-00642-f009:**
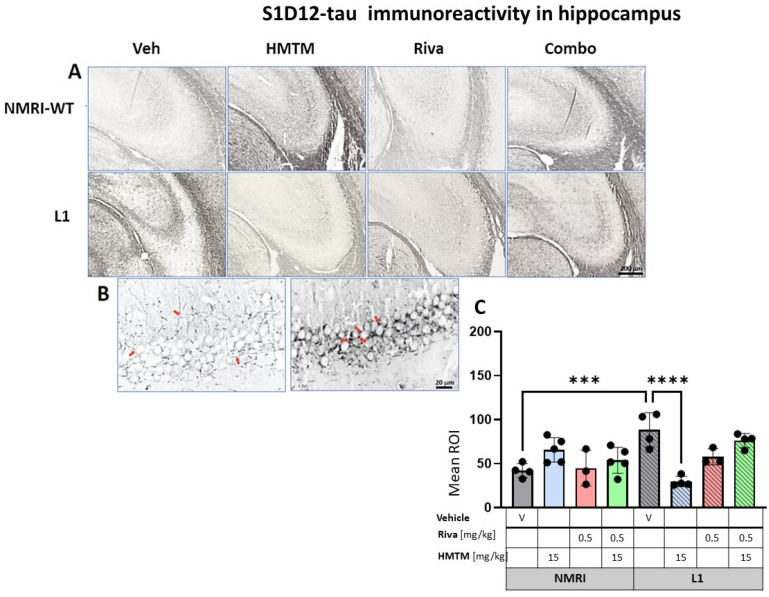
S1D12 tau immunohistochemistry in the hippocampus of wild-type and L1 mice. The core groups were treated as indicated with the vehicle, HMTM, rivastigmine and a combination of the two compounds. (**A**) Representative images of S1D12 immunoreactivity in the hippocampal CA3. (**B**) High magnification of anti-tau hippocampal staining in vehicle-treated NMRI-WT (on the **left**) and L1 (on the **right**) mice showing a different pattern of tau labeling in neural processes in both groups. In WT mice, immunolabeling was seen in thin afferent fiber terminals in the stratum lucidum and fine varicosities adjacent to the pyramidal cell membrane (arrows in the left image). In L1 mice, the presence of dark granular deposits in the vicinity of pyramidal cell surface may indicate elevated tau levels in the synapses (arrows in the right image). (**C**) The mean relative optical density (mean ROI) of S1D12 staining in the hippocampus. An ANOVA of the core groups revealed a significant effect of the genotype, treatment and an interaction between these factors (F’s < 3.9; *p*’s < 0.0026). Two-tailed planned comparisons confirmed a significant increase in tau levels in the hippocampus of L1 mice compared to NMRI (t = 4.4, df = 6; *p* = 0.0005), and this phenotype was recovered upon the administration of HMTM (t = 5.8, df = 6; *p* = 0.001). The rivastigmine treatment and combination of rivastigmine and HMTM did not lead to added benefits. NMRI treated with HMTM showed a significant increase in tau immunoreactivity in comparison with the NMRI vehicle group (t = 3, df = 7; *p* = 0.02). Values presented as the mean ± SD with scatter plots for individual data. Statistical significance between the groups was calculated using the Newman–Keuls test (*** *p* < 0.001, **** *p* < 0.0001). Scale bars: 200 µm for cortex microphotographs and 20 μm for the insertions. For the WT-Veh and WT-Combo groups n = 5; for the L1-Veh and L1-Combo groups n = 4; for the WT-Riva, L1-LMTM and L1-Riva groups n = 3.

**Table 1 cells-13-00642-t001:** The treatment groups, drug doses used and cohort sizes for NMRI wild-type and L1 tau transgenic mice.

Core Groups							
Groups	Genotype	Sex	Age at the start day of the study	Treatment (weeks 1–5)	Treatment (weeks 6–11)	Mice no. at the beginning of the study	Mice no. at the end of the study
WT-Veh	NMRI-WT	F	23–27 weeks	Veh	Veh	9	9
WT-HMTM	NMRI-WT	F		Veh	HMTM 15	10	10
WT-Riva	NMRI-WT	F		Riva 0.5	Riva 0.5	9	8 *
WT-COMBO	NMRI-WT	F		Riva 0.5	Riva 0.5 + HMTM 15	10	10
L1-Veh	NMRI-tgL1	F	19–27 weeks	Veh	Veh	9	9
L1-HMTM	NMRI-tgL1	F		Veh	HMTM 15	9	8 *
L1-Riva	NMRI-tgL1	F		Riva 0.5	Riva 0.5	9	8 *
L1-COMBO	NMRI-tgL1	F		Riva 0.5	Riva 0.5 + HMTM 15	9	9
					Total number of animals	74	71
**Complementary Groups**							
Groups	Genotype	Sex	Age at the start day of the study	Treatment (weeks 1–5)	Treatment (weeks 6–11)	Mice no. at the beginning of the study	Mice no. at the end of the study
L1-HMTM	NMRI-tgL1	F	19–27 weeks	Veh	HMTM 5	9	9
L1-Riva	NMRI-tgL1	F		Riva 0.1	Riva 0.1	9	9
L1-COMBO	NMRI-tgL1	F		Riva 0.1	Riva 0.1 + HMTM 5	9	9
L1-COMBO	NMRI-tgL1	F		Riva 0.1	Riva 0.1 + HMTM 15	9	9
L1-COMBO	NMRI-tgL1	F		Riva 0.5	Riva 0.5 + HMTM 5	9	9
					Total number of animals	45	45

* Two mice died due to poor gavaging tolerance; one on day 38 (L1 treated with HMTM) and a second on day 53 (NMRI-WT treated with Riva). One mouse was euthanized on day 46 (L1 treated with Riva) due to loss of body weight. All drug doses are expressed in mg/kg per day. Veh—vehicle; HMTM—hydroxymethylthionine; Riva—rivastigmine.

**Table 2 cells-13-00642-t002:** List of antibodies, suppliers and dilutions used.

Primary Antibody	Immunogen/ Epitope	Host Species	Dilution	Mono/ Polyclonal	Supplier	Secondary Antibody ^1^	Dilution
Anti-ChAT (Cat. No. #Ab144p)	Human placental ChAT	Goat	1:200	Polyclonal	Merck	Rabbit anti-goat (Cat. No. AP106P)	1:200
Anti-TrkA (Cat. No. #06-574)	Recombinant protein corresponding to the extracellular domain of rat TrkA receptor	Rabbit	1:200	Polyclonal	Merck	Goat anti-rabbit (Cat. No. AP307P)	1:200
Anti-VAChT (Cat. No. 139 103)	Recombinant protein corresponding to residues near the carboxy terminus of rat VAChT	Rabbit	1:500	Polyclonal	Synaptic System	Goat anti-rabbit (Cat. No. AP307P)	1:100
Anti-tau S1D12	Human truncated tau297–391/ epitope: 337–355	Sheep ^2^	1:100	Monoclonal	Genting TauRx Diagnostic Centre	Goat anti-mouse (Cat. No. AP124)	1:100

^1^ Secondary antibodies supplied by Merck. ^2^ S1D12 derived from sheep immunized with tau297–391 and antibody reformatted as a mouse IgG.

## Data Availability

All the raw data supporting the results of this study are available upon the reader’s request.

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
