# Peer review of "Neuroprotection of Cholinergic Neurons with a Tau Aggregation Inhibitor and Rivastigmine in an Alzheimer’s-like Tauopathy Mouse Model"

_cells, 2024, doi:10.3390/cells13070642_

Round 1

Reviewer 1 Report

Comments and Suggestions for Authors

The authors present a very interesting report, which confirms earlier reports, that HMTM, a tau targeting therapeutic, will normalize cholinergic functions in the basal forebrain and projection areas of transgenic mice containing AD-associated tau. And the authors replicate the conundrum wherein concurrent treatment with RIV, a cholinesterase inhibitor used in the symptomatic treatment of AD, blocks the tau reduction induced by HMTM. Insofar as these two could reasonably be used concurrently in the treatment of AD, and were casually used concurrently in the initial clinical trials without the expectation of any antagonistic effects, this is an important effect that deserves thorough study. And an unanswered question is whether or not this interaction can be expected with other cholinesterase inhibitors - not adequately addressed by the authors although their report is related RIV alone. 

The authors do not discuss the special significance of using RIV exclusively except that RIV is the cholinesterase inhibitor of interest because of prior clinical experience. This could be further explored and briefly discussed. RIV has some properties that are not shared by all cholinesterase inhibitors, inhibiting both AChE and BChE, although that difference is likely not important, but should merit a comment. A MAJOR STRENGTH OF THIS REPORT IS THAT RIV WAS USED OVER A LONG PERIOD OF TIME, not just a one-time, or short-term, administration. This improved the value of the report and the importance of the findings.

This report confirms the general hypothesis that tau induces the specific loss of the cholinergic phenotype (measured by ChAT and TrkA), as independent from general neuronal atrophy of basal forebrain neurons. This is an important distinction. And that such loss of cholinergic phenotype can be reversed by targeting AD-associated tau. 

It is reassuring that HMTM will work as indicated against human tau in the mouse model, which by the way is well justified and supported, but the authors do not - BUT SHOULD - comment on how/why HMTM increases the murine tau. Maybe this reviewer missed a comment somewhere, but this is important with regard to mechanism of action - and it may provide insight into the interaction with cholinesterase inhibitors. Please comment if appropriate. 

This is an extensive, well written, well designed and scientifically important report. There are only minor errors in the manuscript that require corrections. These are on line 85, the references should be [26-28], not including a reference to 18, a typo. Also on lines 343-344, a sentence begins with... "In the basal forebrain,..." and correctly states that cholinergic interneurons are mainly in the striatum and nucleus accumbens, but this sentence is confusing. It should, perhaps state that in the "basal forebrain cholinergic system" or simply state " in the CNS"... as it is written, it suggests that short interneurons are found in the basal forebrain itself, and then goes on to name other structures. It is a little confusing and might be clarified.

Overall, an important contribution. 

Author Response

Responses to Reviewer 1 comments:

We are pleased to note the interest that the Reviewer has taken in our study and we would like to thank him for the specific points he has raised.

  1. The authors present a very interesting report, which confirms earlier reports, that HMTM, a tau targeting therapeutic, will normalize cholinergic functions in the basal forebrain and projection areas of transgenic mice containing AD-associated tau. And the authors replicate the conundrum wherein concurrent treatment with RIV, a cholinesterase inhibitor used in the symptomatic treatment of AD, blocks the tau reduction induced by HMTM. Insofar as these two could reasonably be used concurrently in the treatment of AD, and were casually used concurrently in the initial clinical trials without the expectation of any antagonistic effects, this is an important effect that deserves thorough study. And an unanswered question is whether or not this interaction can be expected with other cholinesterase inhibitors - not adequately addressed by the authors although their report is related RIV alone. 

ANSWER:

This is a very interesting comment from the reviewer and something that needs to be investigated in future studies. All symptomatic treatments currently prescribed for Alzheimer's disease (donepezil, rivastigmine, memantine, ...) interfere with HMTM, so similar results might be expected in experimental models for each of these drugs. However, further work is needed to confirm the mechanisms for each drug. This is stated in “Discussion”, lines 778-795.

We have edited the text (you can see track changes in the manuscript) and copied the text here for convenience:

In the clinical trials, participants were frequently being prescribed one of the symptomatic treatments currently approved for Alzheimer's disease (donepezil, rivastigmine, memantine); interference with HMTM might be expected in experimental models with each of these drugs but further work is required to understand the mechanism of action for each drug. Our results suggest that rivastigmine, donepezil and galantamine differ in their pharmacology. While donepezil and galantamine do not inhibit BuChE and are associated with an increase in CSF AChE protein levels, rivastigmine provides a sustained inhibition of AChE and BuChE [80]. However, we have no evidence that this difference in the mechanisms of action of cholinesterase inhibitors is important in reducing the efficacy of add-on therapy with HMTM.

To our knowledge, we are the first to investigate Riva/HMTM combination treatment in tau models of AD at multiple levels/systems of the brain and have demonstrated decreased pharmacological efficacy of combination therapy in the regulation of hippocampal acetylcholine [15], synaptosomal glutamate release and mitochondrial activity [34,81], and presynaptic expression of SNARE proteins involved in synaptic vesicle docking and fusion [82]. Although a decrease in brain MT+ was observed in animals undergoing long-term rivastigmine treatment [80], MT levels were still within the therapeutic range and should have been effective. The block of HMTM activity therefore is not due to decreased bioavailability of…………

We have also made a few other edits elsewhere to replace „reduced effect on HMTM” „with decreased effect” to avoid potential confusion with “chemical reduction” of HMTM.

  1. The authors do not discuss the special significance of using RIV exclusively except that RIV is the cholinesterase inhibitor of interest because of prior clinical experience. This could be further explored and briefly discussed. RIV has some properties that are not shared by all cholinesterase inhibitors, inhibiting both AChE and BChE, although that difference is likely not important, but should merit a comment. A MAJOR STRENGTH OF THIS REPORT IS THAT RIV WAS USED OVER A LONG PERIOD OF TIME, not just a one-time, or short-term, administration. This improved the value of the report and the importance of the findings.

ANSWER:

It is true that there are pharmacological differences between rivastigmine, donepezil and galantamine. Rivastigmine inhibits AChE and BuChE in the long term, whereas donepezil and galantamine do not inhibit BuChE and are associated with increased levels of AChE protein in the cerebrospinal fluid, and their effects are rapidly reversible. Although difficult to establish conclusively in our study, the specificity of rivastigmine action in interacting with HMTM seems rather unimportant, given that a similar interference was observed in phase III clinical trials with donepezil. Nevertheless, we now mention the pharmacological differences in the action of cholinesterase blockers in the discussion, lines 783-789.

  1. This report confirms the general hypothesis that tau induces the specific loss of the cholinergic phenotype (measured by ChAT and TrkA), as independent from general neuronal atrophy of basal forebrain neurons. This is an important distinction. And that such loss of cholinergic phenotype can be reversed by targeting AD-associated tau. 

ANSWER:

The reviewer has raised a very prudent point. It is indeed possible that our animal model, given the complete recovery of the cholinergic system, together with the normalisation of tau levels and cognitive recovery previously reported (Melis et al. 2015), represents early-stage AD or mild cognitive impairment. Similar results have recently been reported for HMTM in the LUCIDITY confirmatory Phase III clinical trial, in which HMTM administered at 16 mg/day led to complete recovery of cognitive function in MCI patients over 6-12 months. These results were reported at the AD/PD conference in March 2024 and will be submitted for full publication. In view of this, we are of the opinion that the inclusion of this issue in the discussion is not advisable at this time. However, we offer alternative explanations for the mechanism of recovery in our discussion section 4.3.

  1. It is reassuring that HMTM will work as indicated against human tau in the mouse model, which by the way is well justified and supported, but the authors do not - BUT SHOULD - comment on how/why HMTM increases the murine tau. Maybe this reviewer missed a comment somewhere, but this is important with regard to mechanism of action - and it may provide insight into the interaction with cholinesterase inhibitors. Please comment if appropriate. 

ANSWER:

From the results obtained with the particular antibodies used in, it is difficult to interpret different actions of HMTM on murine and human tau species. The following points may be relevant: 1. In human patients, the predominant event in response to HMTM would be a reduction in aggregated tau. 2. In our Line 1 mouse model, we have not yet been able to specify which tau species (monomeric, oligomeric, fibrillary) are predominant and which of these species gets decreased by HMTM. 3. A similar argument can be made for murine tau, which is increased after HMTM treatment. Whether this is a compensatory response or a different species to the normally expressed tubulin-binding form remains elusive. We have recently initiated a programme of work to address these specific issues but, in the meantime, would like to avoid any unnecessary speculation at this stage.

  1. This is an extensive, well written, well designed and scientifically important report. There are only minor errors in the manuscript that require corrections. These are on line 85, the references should be [26-28], not including a reference to 18, a typo. Also on lines 343-344, a sentence begins with... "In the basal forebrain,..." and correctly states that cholinergic interneurons are mainly in the striatum and nucleus accumbens, but this sentence is confusing. It should, perhaps state that in the "basal forebrain cholinergic system" or simply state " in the CNS"... as it is written, it suggests that short interneurons are found in the basal forebrain itself, and then goes on to name other structures. It is a little confusing and might be clarified.

ANSWER:

We thank the Reviewer for his careful attention to editorial errors. The corrections have been incorporated into the manuscript using the Track Changes function.

Reviewer 2 Report

Comments and Suggestions for Authors

The study by Zadrozny et al provides further evidence of a decreased cholinergic phenotype in L1 mice due to their developing tau pathology. Further to it, the authors also provide further evidence that HMTM delays or reverses this loss of cholinergic phenotype and that this effect is lost in presence of rivastigmine, as shown in their past reports (doi: 10.1111/jnc.15553), recapitulating the observations in human patients.

The study is of high interest. However, some points must be addressed properly, and the figures must be improved in terms of quality, size, and proofreading.

It is clear that L1 mice represent a good model of cholinergic loss related to tau pathology. Still, the results and discussion related to cell morphology must be revised. The authors refer to a “reduction in the morphometric parameters of ChAT- and TrkA-immunolabeled neurons” (line 748, etc.). However, the size and quality of the micrographs provided do not allow to appreciate differences in cell morphology although differences in immunoreactivity are clear. Also, no morphometric measure is provided. The authors should provide high resolution pictures and enlarge the inserts.

Similarly, the size of the micrographs presented in figure 6-7 is too small to fully appreciate the characteristics discussed in section 3.2.1. Also in Figure 6 and 7, HMTM instead of LMTM.

Figure 8 and 9 are inverted (Fig 8 is hippocampus). Please ensure that the graphs and labels match the pictures.

Essential negative control studies seem to be missing for the immunohistochemical studies related to the newly developed anti-tau antibody S1D12, particularly since the immunoreactive signal seems to be quite strong in some WT mice.

Some statements are not well supported and should be reformulated, for example lines 770-773: “here report reduced pharmacological efficacy of HMTM at multiple brain 770 levels/systems including…”

In their discussion, the authors provide potential explanations on how the cholinergic phenotype may be recovered with HMTM, although the second explanation will certainly be a matter of debate. Still, it is unclear why upregulating cholinergic tone with rivastigmine pretreatment negatively interferes with HTMT action on the rescue of cholinergic cells and lowering of tau levels (lines 810-813). The authors should also discuss this aspect.

Minor comment: Line 378: wild-Table 1. mice.

Author Response

Responses to Reviewer 2 comments:

The study by Zadrozny et al provides further evidence of a decreased cholinergic phenotype in L1 mice due to their developing tau pathology. Further to it, the authors also provide further evidence that HMTM delays or reverses this loss of cholinergic phenotype and that this effect is lost in presence of rivastigmine, as shown in their past reports (doi: 10.1111/jnc.15553), recapitulating the observations in human patients.

The study is of high interest. However, some points must be addressed properly, and the figures must be improved in terms of quality, size, and proofreading.

ANSWER:

We are pleased to note the interest that the Reviewer has taken in our study and we thank him for the specific points raised.

  1. It is clear that L1 mice represent a good model of cholinergic loss related to tau pathology. Still, the results and discussion related to cell morphology must be revised. The authors refer to a “reduction in the morphometric parameters of ChAT- and TrkA-immunolabeled neurons” (line 748, etc.). However, the size and quality of the micrographs provided do not allow to appreciate differences in cell morphology although differences in immunoreactivity are clear. Also, no morphometric measure is provided. The authors should provide high resolution pictures and enlarge the inserts.

  1. Similarly, the size of the micrographs presented in figure 6-7 is too small to fully appreciate the characteristics discussed in section 3.2.1. Also in Figure 6 and 7, HMTM instead of LMTM.

  1. Figure 8 and 9 are inverted (Fig 8 is hippocampus). Please ensure that the graphs and labels match the pictures.

ANSWER:

Revised figures are included in the current version of the manuscript in response to the Reviewer's request. We are very grateful to the Reviewer for his pertinent comments and apologize for not taking sufficient care in preparation of the figures.

  1. Essential negative control studies seem to be missing for the immunohistochemical studies related to the newly developed anti-tau antibody S1D12, particularly since the immunoreactive signal seems to be quite strong in some WT mice.

ANSWER:

This is an important issue, which we have considered a priori. The difficulty however is that (i) we do not have knock-out tissue to confirm the Abs specificity, and (ii) that the epitope in the repeat domain of the tau protein, against which S1D12 is directed is conserved and present in both murine and human tau (and thus labels both wild-type and transgenic tissue, but with clearly distinguishing intensities). In addition, (iii) routine controls without primary or secondary antibody were performed for S1D12 (as for all other antibodies) and no specific immunolabeling was observed in either the positive or the negative controls.

  1. Some statements are not well supported and should be reformulated, for example lines 770-773: “here report reduced pharmacological efficacy of HMTM at multiple brain 770 levels/systems including…”

ANSWER:

We have reworded this statement, and the current version can be found on lines 790- 795.

  1. In their discussion, the authors provide potential explanations on how the cholinergic phenotype may be recovered with HMTM, although the second explanation will certainly be a matter of debate. Still, it is unclear why upregulating cholinergic tone with rivastigmine pretreatment negatively interferes with HTMT action on the rescue of cholinergic cells and lowering of tau levels (lines 810-813). The authors should also discuss this aspect.

ANSWER:

Overall, our two explanations still stand. They can be reduced to: 1. Riva as an AChEI acts at the synaptic level to increase the half-life of released ACh. It is unlikely to restore the cholinergic phenotype. The continued accumulation of tau in neurons is not affected by Riva. Consequently, over time, the cholinergic tone continues to decline (despite high doses of Riva) and the neurons stop releasing ACh. This accurately models the symptomatic therapy of Riva in patients. 2. HMTM has a disease-modifying effect by reducing tau aggregation in cholinergic neurons, increasing cholinergic tone and (when administered alone) increasing AChE activity (Figure 7) to modulate cholinergic levels in the synaptic cleft. In this scenario, the reduction in tau is directly responsible for (not necessarily correlated with) the recovery of cholinergic tone. 3) As an extension of 1 and 2, long-term treatment with rivastigmine further depletes cholinergic neurons and the brain homeostatically adapts to these drug effects. However, the mechanisms by which these homeostatic adjustments occur and how they prevent HMTM from lowering tau in cholinergic projection targets remain elusive.

  1. Minor comment: Line 378: wild-Table 1. mice.

ANSWER:

This has been corrected.

Round 2

Reviewer 2 Report

Comments and Suggestions for Authors

Thank you for addressing the comments.